# Target-to-Source Augmentation for Aspect Sentiment Triplet Extraction

**Yice Zhang**[1,3], **Yifan Yang**[1,2], **Meng Li**[1], **Bin Liang**[1,2,4*], **Shiwei Chen**[1,3], **Ruifeng Xu**[1,2,3*]

[1] Harbin Insitute of Technology, Shenzhen, China
[2] Guangdong Provincial Key Laboratory of Novel Security Intelligence Technologies
[3] Peng Cheng Laboratory, Shenzhen, China
[4] The Chinese University of Hong Kong, Hong Kong, China

{zhangyc_hit,evanyfyang}@163.com
mimas.li777@gmail.com, chenshw@pcl.ac.cn
bin.liang@cuhk.edu.hk, xuruifeng@hit.edu.cn

## Abstract

Aspect Sentiment Triplet Extraction (ASTE) is an important task in sentiment analysis, aiming to extract aspect-level opinions and sentiments from user-generated reviews. The fine-grained nature of ASTE incurs a high annotation cost, while the scarcity of annotated data limits the performance of existing methods. This paper exploits data augmentation to address this issue. Traditional augmentation methods typically modify the input sentences of existing samples via heuristic rules or language models, which have shown success in text classification tasks. However, applying these methods to fine-grained tasks like ASTE poses challenges in generating diverse augmented samples while maintaining alignment between modified sentences and origin labels. Therefore, this paper proposes a target-to-source augmentation approach for ASTE. Our approach focuses on learning a generator that can directly generate new sentences based on labels and syntactic templates. With this generator, we can generate a substantial number of diverse augmented samples by mixing labels and syntactic templates from different samples. Besides, to ensure the quality of the generated sentence, we introduce fluency and alignment discriminators to provide feedback on the generated sentence and then use this feedback to optimize the generator via a reinforcement learning framework. Experiments demonstrate that our approach significantly enhances the performance of existing ASTE models.[1]

## 1 Introduction

Aspect Sentiment Triplet Extraction (ASTE) is an important task in sentiment analysis (Peng et al., 2020), which is receiving increasing attention in

---

*  * Corresponding Authors
[1]We release our code and data at https://github.com/HITSZ-HLT/T2S-Augmentation.

Figure 1: Samples synthesized by traditional augmentation method and the proposed target-to-source method.

natural language processing (Xu et al., 2020; Chen et al., 2021; Yan et al., 2021; Zhang et al., 2021; Xu et al., 2021; Chen et al., 2022; Zhang et al., 2022). The goal of ASTE is to extract aspect-level opinions and sentiments from user-generated reviews. For example, given the review sentence "*the price is reasonable although the service is poor*", the output of ASTE would be {(*price*, *reasonable*, positive), (*service*, *poor*, negative)}.

The ASTE task faces the challenge of data scarcity, and data augmentation has emerged as a potential solution to this issue. Annotating ASTE data is time-consuming and labor-intensive due to its fine-grained nature, leading to small existing public ASTE datasets. Despite some few-shot attempts (Hosseini-Asl et al., 2022; Liang et al., 2023), mainstream methods still require a significant amount of annotated data to achieve satisfactory performance. Data augmentation is the technique that synthesizes new training samples based

on existing datasets, and researchers have explored data augmentation methods to alleviate the problem of data scarcity (Li et al., 2020; Wang et al., 2021; Hsu et al., 2021; Liang et al., 2021; Wang et al., 2022; Hu et al., 2022; Li et al., 2022).

Traditional augmentation methods typically modify the input sentence of the existing sample and then combine the modified sentence with the original label to form an augmented sample (Zhang et al., 2015; Wei and Zou, 2019; Kobayashi, 2018; Wu et al., 2019). While such methods have shown improvements in text classification tasks, ensuring alignment between the modified sentence and the original label in fine-grained tasks like ASTE is challenging. Existing efforts often limit the range or extent of modifications to ensure alignment (Li et al., 2020; Hsu et al., 2021; Gao et al., 2022), but these approaches restrict the diversity of the augmented samples and thereby weaken the effectiveness of data augmentation. Furthermore, for the ASTE task, labels and syntactic structures are two primary sources of diversity. For example, consider the sentences "*the price is reasonable*" and "*it provides an intimate setting*". Firstly, their corresponding labels, (*price*, *reasonable*, positive) and (*setting*, *intimate*, positive), differ. Secondly, the syntactic relations also vary: *reasonable* and *price* exhibit an adjective complement (acomp) relation, whereas *intimate* and *setting* display an adjective modifier (amod) relation.

Based on the above observations, this paper proposes a target-to-source augmentation approach for ASTE. The main idea is to learn a generator that can directly generate new sentences based on both labels and syntactic templates. Compared to traditional augmentation methods, our approach enables the generation of more diverse contexts for a given label by leveraging various syntactic templates. More importantly, we can produce a substantial number of augmented samples by combining labels and syntactic templates from different samples. This allows us to further select high-quality ones from these augmented samples to construct the augmented dataset.

In addition to diversity, augmented samples must also meet two requirements: fluency and alignment. Fluency refers to the ability of the generator to produce natural and coherent sentences. Alignment requires consistency between the given labels and the generated sentences. To satisfy these two requirements, we introduce two discriminators, namely the fluency discriminator and the alignment discriminator, to assess the quality of the generated sentences. Next, we employ a reinforcement learning framework to optimize the generator based on the feedback from these two discriminators. Finally, in the inference phase, we use these two discriminators to further filter out low-quality augmented samples. Experiments demonstrate that our augmentation approach outperforms previous augmentation methods and significantly improves the performance of existing ASTE models.

## 2 Related Work

### 2.1 Data Augmentation

Data augmentation aims at synthesizing new training samples based on existing datasets. Most data augmentation methods typically modify sentences through heuristic rules (Zhang et al., 2015; Wei and Zou, 2019) or language models (Kobayashi, 2018; Wu et al., 2019). The most representative rule-based method is EDA (Wei and Zou, 2019), which involves four operations: synonym replacement, random insertion, random swap, and random deletion. The language model-based methods generally follow the corrupt-and-reconstruct framework, which first randomly masks words or spans in the sentence and then uses a language model to fill the masked parts.

For fine-grained tasks, these methods can easily lead to misalignment between the modified sentences and the original labels. There have been two main attempts to solve this problem:

1. **Label-unrelated Modification**: this method first identifies label-unrelated parts of the sentences through rules or selective perturbed masking and then only modifies these parts to avoid misalignment (Li et al., 2020; Hsu et al., 2021; Gao et al., 2022).

2. **Conditional Language Modeling**: this method inputs both the label sequence and the sentence into the language model (Li et al., 2020) or linearizes the label into the sentence and inputs it into the language model (Ding et al., 2020; Zhou et al., 2022).

Although these attempts can appropriately alleviate the problem of misalignment, the issue still persists due to a lack of supervision over the modified content.

In addition, there are other data augmentation methods such as back-translation (Sennrich et al.,

2016), MixUp (Zhang et al., 2018), and reinforcement learning-guided generation (Liu et al., 2020), but few works apply them to fine-grained tasks.

## 2.2 Data Augmentation for Aspect-Based Sentiment Analysis

Unlike sentence-level sentiment analysis, Aspect-Based Sentiment Analysis (ABSA) focuses on opinions and sentiments expressed on specific aspects within a sentence (Pontiki et al., 2014). Aspect Sentiment Triplet Extraction (ASTE) is a representative task in the current research of ABSA, proposed by Peng et al. (2020).

As fine-grained tasks, ABSA faces the problem of data scarcity, and researchers have developed various data augmentation methods to alleviate this issue. Dai and Song (2019) apply rules to label auxiliary data. Li et al. (2020) propose a conditional data augmentation method for Aspect Term Extraction (ATE). Hsu et al. (2021) adopt selective perturbed masking to select label-unrelated parts of sentences and then use language models to replace these parts. Wang et al. (2021) propose a progressive self-training framework to infer pseudolabels on the unlabeled data. Liang et al. (2021) design aspect-invariant/-dependent data augmentation and deploy a supervised contrastive learning objective. Wang et al. (2022) conduct both aspect augmentation and polarity augmentation. Hu et al. (2022) propose a template-order data augmentation method for generative ABSA.

## 3 Problem Definition

Target-to-source augmentation aims to learn a generator that can generate corresponding sentences based on given labels and syntactic templates. In this paper, we use the dependency tree as the syntactic template. Formally, the generator is defined as follows:

$$\mathbb{G} : L \times D \to S, \tag{1}$$

where $l \in L$ denotes the given label, *i.e.*, a set of aspect-sentiment triplets, $d \in D$ denotes a dependency tree, and $s \in S$ denotes the generated sentence. The generated sentence and the given label together form an augmented sample $(s, l)$.

We have three requirements for the augmented sample:

1. **Fluency**: the generated sentence should be close to real-world sentences in terms of form and structure, without logical or grammatical errors.

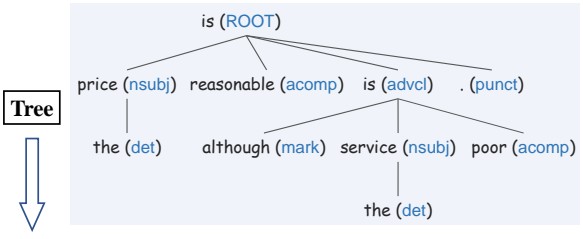

Figure 2: Prefix notation for a dependency tree.

2. **Alignment**: the generated sentence should contain the given aspect-sentiment triplets and not introduce new aspect-level sentiment and opinion.

3. **Diversity**: the augmented sample should have significant differences from existing samples.

## 4 Proposed Framework

### 4.1 Model Architectures

To assist in the training of the generator, our framework includes two additional models: a fluency discriminator and an alignment discriminator. The fluency discriminator assesses the fluency of the generated sentence, and the alignment discriminator evaluates whether the generated sentence aligns with aspect-sentiment triplets. These two models can be formulated as follows:

$$\mathbb{D}_{\text{flu}} : S \to \{0, 1\}, \tag{2}$$
$$\mathbb{D}_{\text{ali}} : S \times L \to \{0, 1\}. \tag{3}$$

Our generator is a complete transformer structure, consisting of an encoder and a decoder. The two discriminators consist of a transformer encoder and a binary classifier.

To fit the transformer model, we need to convert the aspect-sentiment triplets and dependency trees into flattened sequences. For the aspect-sentiment triplets, we use a template to convert each triplet into a sequence and then concatenate multiple sequences to form the label sequence (Zhang et al., 2021), which can be formulated as follows:

$$\text{seq}_{\text{label}} = a_1 \mid o_1 \mid p_1 ; \cdots ; a_n \mid o_n \mid p_n, \tag{4}$$

where $(a_i, o_i, p_i)$ denotes an aspect-sentiment triplet. For the dependency tree, we use prefix notation to represent a tree as a sequence (Lample and Charton, 2020), which is illustrated in Figure 2.

## 4.2 Supervised Learning

Before applying reinforcement learning, it is important to provide the generator with a good initialization. In addition, our reinforcement learning framework also requires two discriminators to provide feedback on the generated sentences. Therefore, prior to reinforcement learning, we train these three models using supervised learning.

**Pseudo-labeled Dataset.** The existing labeled dataset is too small to train a usable generator and reliable discriminators. To overcome this limitation, we construct a pseudo-labeled dataset and train models on it. First, we train an ASTE model ([Zhang et al., 2021](#)) on the existing labeled dataset. Then, we use this model to generate pseudo-labels for the unlabeled data and select samples with high-confidence predictions[2] to construct the pseudo-labeled dataset $\tilde{D} = \{(s_1, d_1, l_1), \cdots\}$.

**Training Generator.** We optimize the generator on the pseudo-labeled dataset through the following loss function:

$$L_{\mathbb{G}} = \frac{1}{N} \sum_{s_i, l_i, d_i \in \tilde{D}} \text{CrossEntropy}(\mathbb{G}(l_i, d_i), s_i), \quad (5)$$

where $N$ denotes the size of $\tilde{D}$.

**Training Fluency Discriminator.** We input both fluent and non-fluent sentences to the fluency discriminator and use the cross-entropy loss to optimize it:

$$L_{\mathbb{D}_{\text{flu}}} = -\frac{1}{N_1} \sum_{s_{\text{f}}} \log \mathbb{D}_{\text{flu}}(s_{\text{f}}) \quad (6)$$
$$-\frac{1}{N_2} \sum_{s_{\text{nf}}} \log(1 - \mathbb{D}_{\text{flu}}(s_{\text{nf}})),$$

where $s_{\text{f}}$ and $s_{\text{nf}}$ represent a fluent sentence and a non-fluent sentence, respectively, and $N_1$ and $N_2$ are the total number of two categories.

We use the existing review sentences as fluent sentences, *i.e.*, $s_{\text{f}} \in \tilde{D}$. For non-fluent sentences, we generate them by mixing labels and dependency trees from different samples, inputting them into the trained generator to generate sentences, and then considering the ones with low fluency as non-

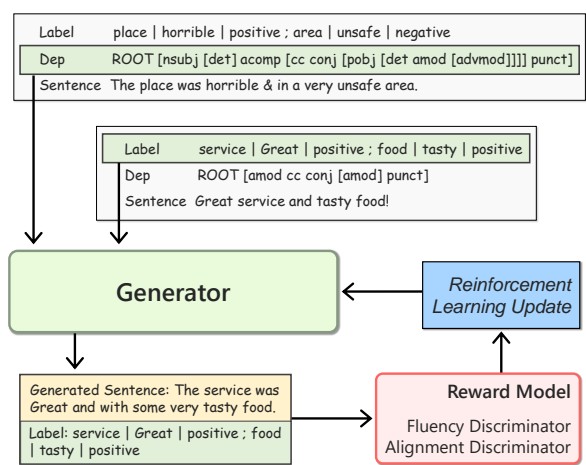

Figure 3: The reinforcement learning framework for tuning a target-to-source generator.

fluent sentences:

$$s_{ij} = \mathbb{G}(l_i, d_j), l_i, d_j \in \tilde{D}. \quad (7)$$
$$D_{\text{nf}} = \{s_{ij} | \text{flu}(s_{ij}) < \text{flu}(s_{jj}) - \theta_{\text{flu}}, \quad (8)$$
$$i, j \in [1, N], i \neq j\},$$

where $\text{flu}(\cdot)$ is a fluency scorer[3] that gives a score from 0 to 1 for the fluency of a sentence, and $\theta_{\text{flu}}$ is a fluency threshold, which we set to 0.1 based on empirical observations.

**Training Alignment Discriminator.** Like the fluency discriminator, we input both aligned and non-aligned samples into the alignment discriminator and use the cross-entropy loss to optimize it:

$$L_{\mathbb{D}_{\text{ali}}} = -\frac{1}{N} \sum_{s, l_{\text{a}} \in \tilde{D}} \log \mathbb{D}_{\text{ali}}(s, l_{\text{a}}) \quad (9)$$
$$-\frac{1}{N} \sum_{s, l_{\text{na}} \in \tilde{D}'} \log(1 - \mathbb{D}_{\text{ali}}(s, l_{\text{na}})),$$

where $(s, l_{\text{a}})$ and $(s, l_{\text{na}})$ represent an aligned sample and a non-aligned sample, respectively. To construct non-aligned samples, we use beam search to generate four prediction results on the trained ASTE model ([Zhang et al., 2021](#)) and then sample one of two predictions with lower confidence as the non-aligned labels $l_{\text{na}}$.

## 4.3 Reinforcement Learning

Supervised learning enables the generator to generate sentences based on labels and dependency

---

[2]Specifically, we exclude pseudo-labels with a confidence level below 0.75, i.e., those satisfying $\exists_t P(y_t | \boldsymbol{y}_{<t}, \boldsymbol{x}) < 0.75$. Refer to Appendix B.1 for the effect of the confidence threshold.

[3]This fluency scorer is from https://github.com/PrithivirajDamodaran/Parrot_Paraphraser. We don't directly use the result of this scorer to determine whether a sentence is fluent or not because real review sentences also have a probability of being scored low by the scorer.

trees. However, when the label and dependency tree come from different samples, the generator tends to generate a non-fluent or non-aligned sentence. To address this problem, we use a reinforcement learning framework to train the generator. As shown in Figure 3, we employ two discriminators to provide feedback on the generated sentence and then optimize the generator based on this feedback.

**Reward Calculation.** Given the label $l_i$ and dependency tree $d_j$, we denote the sentence generated by the generator as $s_{ij}$:

$$s_{ij} = \mathbb{G}^{\mathrm{RL}}(l_i, d_j), i \neq j. \qquad (10)$$

We input the generated sentence $s_{ij}$ into the fluency discriminator to obtain its fluency score $f_{\mathbb{D}_{\mathrm{flu}}}(s_{ij})$ and input both the generated sentence $s_{ij}$ and the given label $l_i$ into the alignment discriminator to obtain its alignment score $f_{\mathbb{D}_{\mathrm{ali}}}(s_{ij}, l_i)$. In addition, to prevent the model from generating short and simple sentences without considering the dependency tree, we also introduce a length penalty $\mathrm{pe}_{\mathrm{len}}(s_{ij}, d_j)$. The reward is a combination of these three parts:

$$\begin{aligned} r(s_{ij}, l_i, d_j) = & f_{\mathbb{D}_{\mathrm{flu}}}(s_{ij}) + f_{\mathbb{D}_{\mathrm{ali}}}(s_{ij}, l_i) \qquad (11) \\ & - \mathrm{pe}_{\mathrm{len}}(s_{ij}, d_j), \end{aligned}$$

where the length penalty is calculated by:

$$\mathrm{pe}_{\mathrm{len}}(s, d) = \frac{\max(\mathrm{len}(d) - \mathrm{len}(s), 0)^2}{\mathrm{len}(d)}. \qquad (12)$$

**Reinforcement Learning Update.** Inspired by Ouyang et al. (2022), we use the Proximal Policy Optimization (PPO) algorithm (Schulman et al., 2017) to perform reinforcement learning updates. We update the generator by maximizing the following objective function:

$$J = E_{(s,l,d) \sim D_{\mathbb{G}^{\mathrm{RL}}}}[r(s, l, d) - \lambda_{\mathrm{KL}} \mathrm{KL}(s, l, d)],$$

where $\lambda_{\mathrm{KL}}$ is a hyper-parameter which we empirically set to 0.01, and the $\mathrm{KL}(s, l, d)$ is a per-token KL penalty from the SL (supervised learning) generator $\mathbb{G}^{\mathrm{SL}}$ at each token which is used to mitigate over-optimization from the discriminators (Ouyang et al., 2022):

$$\mathrm{KL}(s, l, d) = \log(\mathbb{G}^{\mathrm{RL}}(s|l, d) / \mathbb{G}^{\mathrm{SL}}(s|l, d)).$$

### 4.4 Data Synthesis and Filtering

We randomly select two samples from the pseudo-labeled dataset $\tilde{D}$. To guarantee the quality of the

| Dataset | #Sent | #Triplet | #Pos | #Neu | #Neg |
|---|---|---|---|---|---|
| Rest-14-train | 1266 | 2338 | 1692 | 166 | 480 |
| Rest-14-dev | 310 | 577 | 404 | 54 | 119 |
| Rest-14-test | 492 | 994 | 773 | 66 | 155 |
| Lap-14-train | 906 | 1460 | 817 | 126 | 517 |
| Lap-14-dev | 219 | 346 | 169 | 36 | 141 |
| Lap-14-test | 328 | 543 | 364 | 63 | 116 |
| Yelp | 427300 | 602394 | 400858 | 65388 | 136148 |
| Amazon | 188796 | 230265 | 129324 | 18573 | 82368 |

Table 1: Statistics of four ASTE datasets (Xu et al., 2020). #Sent and #Triplet represent the number of sentences and triplets, respectively. Besides, #Pos, #Neu, and #Neg represent the numbers of positive, neutral, and negative triplets, respectively.

generated sentence, we require them to contain an equal number of triplets. Next, we input the label of the first sample and the dependency tree of the second sample into the generator $\mathbb{G}^{\mathrm{RL}}$ to generate the corresponding sentence. This process is repeated $n_1$ times, resulting in $n_1$ augmented samples.

Subsequently, we employ the fluency discriminator and the alignment discriminator to filter out non-fluent or non-aligned samples. The remaining $n_2$ samples will be used to build the augmented dataset.

## 5 Experiment

### 5.1 Datasets

We evaluate the proposed augmentation method on `Restaurant-14` and `Laptop-14` of ASTE-Data-V2 (Xu et al., 2020). Accordingly, we utilize the Yelp Dataset[4] for `Restaurant-14` and Amazon Review Dataset[5] (Ni et al., 2019) for `Laptop-14` to construct pseudo-labeled datasets. For each domain, we utilize a total of 10,000 reviews. We infer the aspect-sentiment triplets for each sentence of the reviews using a pre-trained ASTE model (Zhang et al., 2021) and retain only those sentences that contained triplets. The data statistics for these datasets are shown in Table 1.

### 5.2 Implementation Details

We obtain the dependency tree for each sentence using `spaCy`[6]. Both the generator and the two dis-

---

[4]https://www.yelp.com/dataset
[5]https://nijianmo.github.io/amazon/index.html
[6]The trained pipeline we use is `en_core_web_sm` 3.3.0.

| ASTE Model | Data Augmentation Method | Restaurant-14 | | | | Laptop-14 | | | |
|---|---|---|---|---|---|---|---|---|---|
| | | P. | R. | $F_1$ | $\Delta$-$F_1$ | P. | R. | $F_1$ | $\Delta$-$F_1$ |
| BARTABSA (Yan et al., 2021) | No Augmentation | 68.73 | 67.15 | 67.92 | - | 61.67 | 55.30 | 58.31 | - |
| | EDA(Wei and Zou, 2019) | 67.71 | 66.72 | 67.20 | -0.72 | 55.10 | 50.69 | 52.79 | -5.52 |
| | Mask-then-Fill(Gao et al., 2022) | 65.48 | 63.58 | 64.50 | -3.42 | 55.12 | 49.28 | 52.03 | -6.28 |
| | Conditional-Augmentation-1(Li et al., 2020) | 70.20 | 67.98 | 69.06 | +1.14 | 59.11 | 52.75 | 55.74 | -2.57 |
| | Conditional-Augmentation-2(Li et al., 2020) | 71.23 | 68.30 | 69.72 | +1.80 | 59.41 | 54.53 | 56.84 | -1.47 |
| | **Target-to-Source Augmentation** (Ours) | **74.08** | **72.65** | **73.35** | **+5.43** | **63.84** | **59.15** | **61.39** | **+3.08** |
| GAS (Zhang et al., 2021) | No Augmentation | 73.50 | 71.41 | 72.44 | - | 65.57 | 60.22 | 62.78 | - |
| | EDA(Wei and Zou, 2019) | 70.70 | 70.24 | 70.47 | -1.97 | 62.86 | 57.05 | 59.81 | -2.97 |
| | Mask-then-Fill(Gao et al., 2022) | 68.70 | 66.88 | 67.77 | -4.67 | 60.43 | 53.33 | 56.65 | -6.13 |
| | Conditional-Augmentation-1(Li et al., 2020) | 72.50 | 70.81 | 71.64 | -0.80 | 64.26 | 58.93 | 61.47 | -1.31 |
| | Conditional-Augmentation-2(Li et al., 2020) | 73.58 | 70.65 | 72.08 | -0.36 | 64.06 | 58.82 | 61.33 | -1.45 |
| | **Target-to-Source Augmentation** (Ours) | **75.65** | **74.57** | **75.11** | **+2.67** | **66.17** | **61.95** | **63.99** | **+1.21** |
| Span-ASTE (Xu et al., 2021) | No Augmentation | 71.50 | 71.22 | 71.36 | - | 62.63 | 54.09 | 58.03 | - |
| | EDA(Wei and Zou, 2019) | 71.02 | 66.50 | 68.63 | -2.73 | 65.50 | 49.38 | 56.04 | -1.99 |
| | Mask-then-Fill(Gao et al., 2022) | 66.90 | 63.04 | 64.90 | -6.46 | 53.77 | 46.36 | 49.78 | -8.25 |
| | Conditional-Augmentation-1(Li et al., 2020) | 70.60 | 70.85 | 70.71 | -0.65 | 60.20 | 53.09 | 56.36 | -1.67 |
| | Conditional-Augmentation-2(Li et al., 2020) | 69.96 | 69.87 | 69.90 | -1.46 | 60.56 | 53.94 | 57.04 | -0.99 |
| | **Target-to-Source Augmentation** (Ours) | **72.77** | **74.18** | **73.45** | **+2.09** | **62.99** | **60.04** | **61.46** | **+3.43** |
| BDTF (Zhang et al., 2022) | No Augmentation | 76.29 | 72.43 | 74.30 | - | 65.72 | 56.41 | 60.66 | - |
| | EDA(Wei and Zou, 2019) | 78.05 | 68.11 | 72.66 | -1.64 | 70.85 | 51.13 | 59.29 | -1.37 |
| | Mask-then-Fill(Gao et al., 2022) | 73.73 | 61.55 | 67.02 | -7.28 | 62.25 | 45.58 | 52.43 | -8.23 |
| | Conditional-Augmentation-1(Li et al., 2020) | 78.57 | 68.53 | 73.20 | -1.10 | 71.39 | 54.13 | 60.23 | -0.43 |
| | Conditional-Augmentation-2(Li et al., 2020) | 77.29 | 70.22 | 73.53 | -0.77 | 69.59 | 52.35 | 59.62 | -1.04 |
| | **Target-to-Source Augmentation** (Ours) | **78.15** | **74.97** | **76.52** | **+2.22** | **67.48** | **58.11** | **62.43** | **+1.77** |

Table 2: Experimental results on ASTE-Data-v2 (Xu et al., 2020) (%).

criminators are initialized using T5-large (Raffel et al., 2020). In the PPO algorithm, we set $\lambda = 0.95$ and $\gamma = 0.99$. To harness its effectiveness, we employ several training strategies, including reward clipping, reward scaling, advantage normalization, and learning rate decay. For each domain, we generate 100,000 augmented samples and select 5,000 samples based on alignment and fluency scores. These selected samples are then merged with the previous dataset to create an augmented dataset. We evaluate these augmented datasets using the existing ASTE models. To minimize the impact of randomness, we run each model 5 times with different random seeds and then report the average results.

### 5.3 Baselines

Currently, there are no data augmentation methods specifically designed for the ASTE task. Therefore, we select three data augmentation methods from similar tasks as baselines for comparison.

**EDA** (Wei and Zou, 2019) is a classic rule-based augmentation method, which involves four operations: synonym replacement, random insertion, random swap, and random deletion.

**Mask-then-Fill** (Gao et al., 2022) first randomly masks out a sentence fragment and then infills a variable-length text span with a fine-tuned infilling model. To preserve the sentence label, this method only masks label-unrelated sentence fragments.

**Conditional Augmentation** (Li et al., 2020) is similar to mask-then-fill but with one key difference. Conditional augmentation additionally includes the label as a sequence input to the infilling model. This method is proposed specifically for Aspect Term Extraction, where the label can be easily converted into a sequence of the same length as the input sentence. However, this method cannot be directly applied to the ASTE task, as its label is more fine-grained. We devise two solutions to address this issue: (1) conditional-augmentation-1 only takes the aspect and opinion label as the condition; (2) conditional-augmentation-2 concatenates the triplets and the sentence together as input to the infilling model.

Moreover, we select four representative ASTE models to evaluate the augmented datasets.

**BARTABSA** (Yan et al., 2021) and **GAS** (Zhang et al., 2021) are two generative approaches.

BARTABSA transforms ASTE into an index generation problem and utilizes the BART (Lewis et al., 2020) model to address it. GAS transforms ASTE into a text generation problem and employs the T5 (Raffel et al., 2020) model to solve it.

**Span-ASTE** (Xu et al., 2021) performs term extraction and relation classification through the shared span representations. Additionally, this approach introduces a dual-channel span pruning strategy to mitigate the high computational cost caused by span enumeration.

**BDTF** (Zhang et al., 2022) represents each triplet as a relation region in the 2D table and transforms the ASTE task into the detection and classification of these relation regions.

### 5.4 Main Results

Table 2 lists the comparison results between the proposed augmentation approach and the previous augmentation methods. It can be observed that the previous methods have failed on the ASTE task. Instead of improving performance, they have resulted in varying degrees of performance degradation. Particularly, the mask-then-fill method has led to a performance drop of over 3%. Despite incorporating labels as conditions, the conditional augmentation method still fails to achieve performance improvements in most cases. These observations indicate that ensuring alignment between the modified sentences and the original labels is challenging for fine-grained tasks like ASTE. Without explicit supervision of alignment, models are prone to generating misaligned augmented samples, which will ultimately harm the model's performance.

In contrast, our approach consistently achieves improvements on both datasets, demonstrating the effectiveness of our approach. Compared to having no augmentation, our approach yields performance improvements ranging from 1.21% to 5.43%. These improvements are particularly evident for models that initially have poorer performance.

### 5.5 Evaluation of Discriminators

We conduct human evaluations on small data batches to evaluate the fluency and alignment discriminators. We choose 100 samples from the augmented samples synthesized by the generator. Of these, 50 are classified as fluent by the fluency discriminator, and the remaining 50 are classified as non-fluent. These samples are then shuffled, and a

| Discriminator | Score | Restaurant-14 | Laptop-14 |
|---|---|---|---|
| Fluency | high | 90.00 | 86.67 |
| | medium | 65.00 | 52.50 |
| | low | 83.33 | 73.33 |
| Alignment | high | 86.67 | 80.00 |
| | medium | 65.00 | 62.50 |
| | low | 100.00 | 86.67 |

Table 3: Human evaluation of discriminators (accuracy, %).

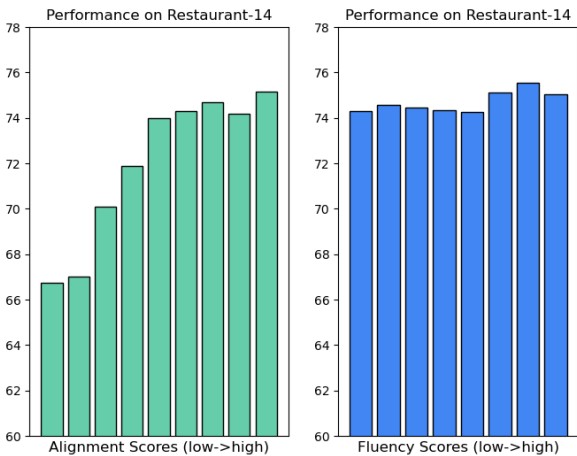

Figure 4: Performance of GAS on the augmented dataset under different alignment and fluency scores ($F_1$-score, %)

human annotator labels each sentence as fluent or non-fluent. We exploit the same method to evaluate the alignment discriminator. As shown in Table 3, these two discriminators generally align with human judgment.

### 5.6 Further Analysis

**Effect of Alignment and Fluency.** We conduct further experiments to analyze the impact of alignment and fluency on the performance of the augmented dataset. From the generated large pool of augmented samples, we perform sample selection based on alignment and fluency scores, ranging from low to high. Subsequently, we evaluate the performance of the corresponding augmented datasets and present the results in Figure 4.

These results indicate that both alignment scores and fluency scores have a positive impact on the performance of the augmented dataset, with higher scores generally leading to better performance. Among these two scores, the alignment scores have a particularly significant influence, while the impact of fluency scores is relatively mild. This finding is somewhat unexpected, suggesting that al-

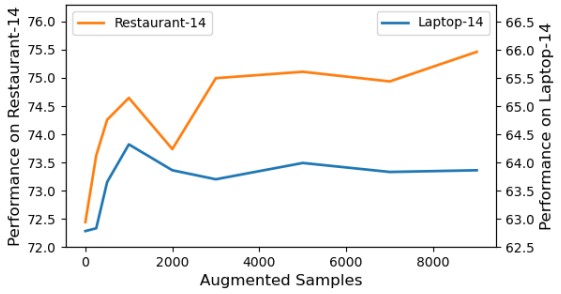

Figure 5: Performance of GAS on the augmented datasets under different numbers of augmented samples ($F_1$-score, %).

| Dataset | Restaurant-14 | Laptop-14 |
|---|---|---|
| No Augmentation | 72.44 | 62.78 |
| Self-training | 74.47 | 62.69 |
| Target-to-Source | **75.11** | **63.99** |
| *w/o reinforcement learning* | 74.24 | 60.70 |
| *w/o new label&syn* | 73.39 | 62.31 |

Table 4: Ablation study ($F_1$-score, %). Self-training refers to using pseudo-labeled data as augmented samples.

though non-fluent sentences may be unfriendly to human readers, their influence on model training is limited.

**Effect of Quantity of Augmented Samples.** The quantity of augmented samples is also a crucial factor in the effectiveness of data augmentation. Increasing the number of augmented samples provides the model with a more diverse range of contexts, thereby enhancing the model's generalization ability. The results depicted in Figure 5 confirm the above statement. As the data quantity increases, the model's performance gradually improves. However, once the data quantity exceeds 3000, the additional improvements become marginal. This observation suggests that the diversity introduced by augmented samples is still limited. Samples with high alignment scores often exhibit a relatively narrow dependency between aspect and opinion, while more complex dependencies can result in looser alignment. Further exploration of this challenge is warranted in further research.

**Ablation Study.** We conduct ablation experiments to analyze the effectiveness of each module. As shown in Table 4, removing reinforcement learning results in a significant performance decrease. Furthermore, when generating augmented samples using only the labels and syntactic templates from the original training set, we observe a performance

drop as well. This highlights the indispensability of these components in the augmentation process.

In addition, we explore the possibility of using the pseudo-labeled dataset as a substitute for the generated augmented samples. The experimental results in Table 4 demonstrate the potential of this approach. Clearly, the pseudo-labeled dataset itself can bring greater diversity. However, the quality of pseudo-labels remains a major concern.

## 5.7 Case Study

Table 5 presents several representative examples to provide an intuitive understanding of the proposed augmentation approach. Observe that the mask-then-fill method easily introduces additional aspect terms or opinion terms, leading to misaligned samples. In comparison, this issue is less severe in conditional augmentation. Furthermore, both methods exhibit limited modifications to the original sentences, resulting in a lack of diversity in the modified sentences. Compared to these methods, our approach generally generates sentences with higher alignment and diversity. Moreover, as our approach can generate numerous sentences by introducing different syntactic templates, even if low-aligned samples are generated, we can easily filter them out using alignment scores. Such an advantage ensures the quality of the augmented dataset.

## 6 Conclusion

In this paper, we propose a target-to-source augmentation approach to alleviate the issue of data scarcity in Aspect Sentiment Triplet Extraction (ASTE). Unlike traditional augmentation methods that modify input sentences, the target-to-source approach focuses on learning a generator to directly generate new sentences based on labels and syntactic templates. By combining the labels and syntactic templates from different samples, this generator can generate a substantial number of diverse augmented samples. To train the generator, we employ a reinforcement learning framework. Specifically, we introduce a fluency discriminator and an alignment discriminator to provide feedback for the generated sentences. Then, we utilize the Proximal Policy Optimization (PPO) algorithm to optimize the generator based on this feedback. We apply the proposed augmentation approach to public ASTE datasets and evaluate the augmented datasets using four representative ASTE models. The experimental results show that the proposed

| Augmentation Methods | Review Sentence | Alignment Scores |
|---|---|---|
| Original Sentence | *The place is so cool and the service is prompt and curtious.* | - |
| Mask-then-Fill | *The place is clean and cool and the service is prompt and curtious.* | -2.36 |
| Conditional Augmentation | *The place is very cool and the service is prompt and curtious.* | 4.63 |
| Target-to-Source Augmentation (1) | *In the end we chose a cool place w/ prompt and curtious service.* | 3.73 |
| Target-to-Source Augmentation (2) | *The place is very cool, the service staff was prompt, and the service was curtious, we looked forward to ordering our food after our meal.* | -0.31 |
| Original Sentence | *The menu is very limited - i think we counted 4 or 5 entrees.* | - |
| Mask-then-Fill | *The menu is very limited and expensive.* | -2.59 |
| Conditional Augmentation | *The menu is very limited - you can only get a couple of the main entrees.* | 4.25 |
| Target-to-Source Augmentation (1) | *I arrived at 7:30 and found myself, with such a limited menu in the corner, I had to do a double take.* | 4.34 |
| Target-to-Source Augmentation (2) | *First off, if you have a craving for scotch eggs, this is the place for you - but be aware that the menu is limited.* | 1.72 |
| Original Sentence | *Unfortunately, the food is outstanding, but everything else about this restaurant is the pits.* | - |
| Mask-then-Fill | *Unfortunately, the food is outstanding and delicious...* | -2.59 |
| Conditional Augmentation | *Unfortunately, the food is outstanding, but the service is a bit of pits.* | -1.63 |
| Target-to-Source Augmentation (1) | *The food at Elmer's is outstanding.* | 3.84 |
| Target-to-Source Augmentation (2) | *We came for outstanding food so if you ever visit Portland, you should consider heading to Sava.* | 5.44 |

Table 5: Case Study. The aspect terms and the opinion terms are marked with blue and red, respectively. In Mask-then-Fill and Conditional Augmentation, the modified parts are marked using underlines.

augmentation approach significantly outperforms the previous augmentation methods and greatly improves the performance of existing ASTE models, demonstrating its effectiveness.

## Acknowledgments

We thank the anonymous reviewers for their valuable suggestions to improve the quality of this work. This work was partially supported by the National Natural Science Foundation of China (62006062, 62176076), Natural Science Foundation of GuangDong 2023A1515012922, Key Technologies Research and Development Program of Shenzhen JSGG20210802154400001, Shenzhen Foundational Research Funding JCYJ20220818102415032, Guangdong Provincial Key Laboratory of Novel Security Intelligence Technologies 2022B1212010005.

## Limitations

Although our approach significantly improves the performance of existing ASTE models, it also suffers from the following limitations:

- Our approach relies on two discriminators to evaluate the quality of the augmented samples. However, these two discriminators are trained on pseudo-labeled datasets, which will inevitably introduce noise. This noise can have a negative impact on data augmentation.

- Our approach achieves diversity by introducing different syntactic templates for each label. However, when the syntactic template does not match the given label, the generator, even after reinforcement learning tuning, tends to produce a significant proportion of incoherent and hard-to-understand sentences. Although we can filter these out, we still have to generate a large number of samples for filtering. Future work can explore how to better introduce diversity.

- Augmented samples inevitably contain noises. Therefore, to better utilize the augmented samples, it is necessary to develop a noise-insensitive training method, which is not addressed in this paper. This is an area that can be explored in future research.

We believe that addressing the above limitations can further enhance the development of data augmentation techniques.

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

## A Discussion on Augmentation Using ChatGPT

As shown in Table 7, utilizing ChatGPT to generate sentences based on labels and dependency trees is feasible, and GPT-4 has demonstrated promising capabilities in this regard. For simpler syntactic structures, GPT-4 can generate sentences that adhere to the requirements. However, when faced with more complex syntactic structures, the output from GPT-4 becomes less predictable and may easily yield misalignment with the given labels.

## B Discussion on Noise

Pseudo-labeling inevitably introduces noise, which is indeed a noteworthy limitation of our approach. To mitigate the potential influence of this noise, we have devised the following strategies:

- We remove samples with invalid labels, including aspect terms and opinion terms not present in the sentence, along with label sequences that are structurally invalid.

- We implement further filtration based on the model's own confidence scores to exclude labels with low confidence.

Based on our observations, these two steps collectively contribute to a significant reduction in noise levels.

Subsequently, we discuss the effects of noise on the generator and alignment discriminator.

- Concerning the generator, it's important to underscore that supervised training with pseudo-labels merely equips it with preliminary sentence generation capabilities. We refine label-sentence alignment through the subsequent reinforcement learning tuning, which heavily relies on the performance of the alignment discriminator.

- Turning to the discriminator, we evaluate its performance on the original training set and find it achieves an impressive accuracy of 89%. While not perfect, the discriminator can identify misalignment in the majority of cases. Importantly, our observations indicate that while borderline cases might exhibit some misalignment, samples with higher alignment scores rarely demonstrate such issues.

In summary, we employ filtering strategies to mitigate noise impact, effectively reducing noise to an acceptable level. Furthermore, the influence of noise on our framework remains limited.

### B.1 Effect of Confidence Threshold

| Confidence Threshold | Alignment Accuracy |
|:---:|:---:|
| 0.7 | 88.20 |
| 0.8 | 88.73 |
| 0.9 | 88.03 |

Table 6: Effect of confidence threshold on the performance of the alignment discriminator on Restaurant-14 (accuracy, %). We evaluate the performance of the alignment discriminator based on its performance on the original training set.

We have discovered that variations in the confidence threshold yield a relatively marginal impact on the alignment discriminator, as evidenced by Table 6. Furthermore, we can reasonably infer that the influence of this confidence threshold on subsequent reinforcement learning tuning is also limited.

| | |
|---|---|
| **Label** | dish \| non-tasty \| negative |
| **Dependency Tree** | ROOT [det prep [pobj] cc conj [det prep [pobj]] prep [pobj [nmod]] punct] |
| **GPT-4** | The dish was non-tasty with bland flavors and lacked any appealing presentation. |
| **Our Approach** | A non-tasty dish and a lot of it for $80. |
| **Label** | server \| jerk \| negative |
| **Dependency Tree** | ROOT [nsubj advmod dobj [det] cc conj [aux xcomp [advmod]] punct punct punct] |
| **GPT-4** | The server was a jerk, not even trying. |
| **Our Approach** | I just had a jerk server and will be staying away!!! |
| **Label** | staff \| unprofessional \| negative ; staff \| shady \| negative |
| **Dependency Tree** | ROOT [prep [prep [pobj [det compound] cc conj [pobj [nmod [det nmod prep [pobj]] compound [amod]]]]] nsubj [det] aux aux attr [neg amod quantmod quantmod] punct] |
| **GPT-4** | Amidst the unprofessional behavior of some staff members, the atmosphere turned shady and the overall experience was not at all pleasant. |
| **Our Approach** | Based on the unprofessional and shady staff at the Orlando Inn of Fashion high end resort this place should have been no more than 4 stars. |
| **Label** | service \| bad \| negative ; manager \| unprofessional \| negative |
| **Dependency Tree** | ROOT [nsubj [det relcl [nsubj advmod prep [pobj]]] acomp [advmod] cc conj [nsubjpass [det compound] auxpass] punct] |
| **GPT-4** | The service, whose manager was unprofessional, was bad and the staff, including the manager, were criticized. |
| **Our Approach** | The service we received here at teh restaurant was really bad and the manager I was assigned was unprofessional. |

Table 7: Several examples of using GPT-4 for target-to-source augmentation.