# OpenReview forum: "Target-to-Source Augmentation for Aspect Sentiment Triplet Extraction"
_EMNLP/2023/Conference — EMNLP 2023 Main_

### Official Review · Reviewer_NQZC · 2023-08-02

**Soundness:** 3

**Excitement:**

3: Ambivalent: It has merits (e.g., it reports state-of-the-art results, the idea is nice), but there are key weaknesses (e.g., it describes incremental work), and it can significantly benefit from another round of revision. However, I won't object to accepting it if my co-reviewers champion it.

**Paper Topic And Main Contributions:**

This paper proposes a novel data augmentation method for Aspect Sentiment Triplet Extraction (ASTE), where they focus on learning a generator that can directly generate new sentences based on labels and syntactic templates.  To ensure the quality of the generated sentence, they introduce fluency and alignment discriminators to provide feedback on the generated sentence and then use the feedback to optimize the generator via a reinforcement learning framework. Extensive experiments show this approach improves the performance of several ASTE models.

**Reasons To Accept:**

1. This paper is clearly written and easy to follow.
2. The experiments involves several representative ASTE models and compare with several popular data augmentation baselines, to show the efficacy of the method.
3. The idea is simple and effective.

**Reasons To Reject:**

1. Limited benchmark datasets. This paper only performs experiments on two ASTE benchmarks. Maybe adding 15Rest and 16Rest of ASTE-Data-v2 would be better to show the approach is effective.
2. Baseline selection. It is unclear how the authors choose the data augmentation baselines. Since ASTE includes aspect term extraction and opinion term extraction which are sequence labelling tasks. Readers might be curious whether authors compare the proposed method with some general sequence labelling-related data augmentation such as DAGA (Ding et al.,).


**Reproducibility:**

3: Could reproduce the results with some difficulty. The settings of parameters are underspecified or subjectively determined; the training/evaluation data are not widely available.

**Reviewer Confidence:**

3: Pretty sure, but there's a chance I missed something. Although I have a good feel for this area in general, I did not carefully check the paper's details, e.g., the math, experimental design, or novelty.

---

> ### Author Rebuttal · Authors · 2023-08-27
>
> We would like to express our great appreciation for your valuable comments to improve the quality of this manuscript.
>
> **Weakness-1: Limited Benchmark Datasets.**
>
> Response-1: Thank you for your valuable feedback. We acknowledge including experimental results on 15Rest and 16Rest can further demonstrate the effectiveness of our approach. We are grateful for your suggestion and will refine this manuscript by incorporating the results of these two datasets.
>
> **Weakness-2: Baseline Selection.**
>
> Response-2: We sincerely appreciate your feedback and thoughtful suggestions. As there are currently no data augmentation methods specifically designed for the ASTE task, we replicate three typical methods from other tasks for comparison. These three methods correspond to traditional rule-based methods, label-unrelated modification methods, and conditional language modeling methods. We appreciate your kind reminder on this matter and apologize for missing out on other methods. In response to this, we will refine this manuscript by providing an explanation for our rationale behind selecting these baselines and including additional baselines, such as DAGA.
>
> Finally, we want to express my sincere gratitude for your insightful and constructive feedback once again. Your input has played a crucial role in improving our work, and we deeply value your thoughtful involvement.

---

### Official Review · Reviewer_Xree · 2023-08-04

**Soundness:** 4

**Excitement:**

4: Strong: This paper deepens the understanding of some phenomenon or lowers the barriers to an existing research direction.

**Paper Topic And Main Contributions:**

This paper proposes a novel way of augmenting data based on labels and syntactic templates to enhance the task of aspect sentiment triplet extraction. The augmentation is achieved by a generator which takes labels and a dependency template as input, and trained via reinforcement learning based on two other discriminators spanning over fluency and alignment scores.

Extensive experiments have been conducted to demonstrate consistent performance gains across different datasets and extraction methods. And additional insights have been provided that showcased the influence of the fluency and alignment scores on the final performance.

**Questions For The Authors:**

1. The alignment discriminator is trained by composing aligned and disaligned examples based on beam search and confidence scores. I wonder what is the beam size and the threshold confidence scores for selecting the negative samples. And how do you decide on these values?
2. Fig 5 shows the trend by changing number of augmentations. What about changing the confidence thresholds?
3. Fig 4 indicates the trends with both scores. Despite the scores given by two models, how about the actual fluency and alignment scores? Even a small set with human-evaluated scores on fluency and alignment is beneficial to see whether the discriminators actually align with humans' perceptions.

**Reasons To Accept:**

1. The idea of automatically training a generator for augmenting the ASTE datasets is inspiring and proved to be effective.
2. The experimental results are promising, and have the potential to be adopted in other tasks requiring tedious annotation schemas.
3. Extensive demonstrations have been shown to prove the effect of the generator trained over different criteria.

**Reasons To Reject:**

It would be even more interesting to see how this method could benefit other structured prediction tasks such as information extraction.


**Reproducibility:**

4: Could mostly reproduce the results, but there may be some variation because of sample variance or minor variations in their interpretation of the protocol or method.

**Reviewer Confidence:**

4: Quite sure. I tried to check the important points carefully. It's unlikely, though conceivable, that I missed something that should affect my ratings.

---

> ### Author Rebuttal · Authors · 2023-08-28
>
> We deeply appreciate your invaluable feedback aimed at enhancing the quality of this manuscript.
>
> **Q1: Details of data construction for training the alignment discriminator.**
>
> A1: Thank you for your valuable feedback. We apologize for any lack of clarity in our manuscript and are committed to refining it for improved comprehensibility.
>
> We train our generator and discriminators on a pseudo-labeled dataset. To ensure the reliability of these pseudo-labels, we retain only those pseudo-labeled samples with high confidence. Specifically, we exclude labels with *minimum-confidence* below 0.75, i.e., those satisfying $\min P(y_t|y_{<t},x)<0.75$. Here, we empirically choose 0.75 as the confidence threshold, and we observe that our approach is not particularly sensitive to this threshold, as illustrated in *A2*.
>
> In training of the alignment discriminator, for a given sample $(x,y)$ in the pseudo-labeled dataset, we utilize the ASTE model with beam search to generate four pseudo-labels, ordered by confidence as $(y_1,y_2,y_3,y_4)$, where typically $y_1=y$. We treat $(x,y_1)$ as the aligned sample, i.e., the positive sample; we treat $(x,y_3)$ and $(x,y_4)$ as non-aligned samples, i.e., negative samples. In the above process, we set the beam size to 4, because a larger beam size would generate obviously unreasonable negative samples; we do not treat $y_2$ as a negative sample to reduce potential noise.
>
>
> **Q2: The impact of the confidence threshold.**
>
> A2: We extend our appreciation for your insightful comments. We have discovered that variations in the confidence threshold yield a relatively marginal impact on the alignment discriminator, as evidenced by the following table. Furthermore, we can reasonably infer that the influence of this confidence threshold on subsequent reinforcement learning tuning is also limited. We thank you for highlighting this aspect, and we will incorporate this analysis to further enhance the manuscript.
>
> | Confidence Threshold | Alignment Accuracy in Training Set |
> | :---------------: | :------: |
> | 0.6 | 0.8820 |
> | 0.7 | 0.8820 |
> | 0.8 | 0.8873 |
> | 0.9 | 0.8803 |
>
> **Q3: Human evaluation for the fluency and alignment discriminators**
>
> A3: Thank you for your constructive suggestions. Given the critical role of these two discriminators in our approach, we agree on the necessity of subjecting them to human evaluation. We will incorporate this into the manuscript.
>
> Finally, we want to convey my sincere gratitude for your insightful and constructive feedback once again. Your contributions have been instrumental in the enhancement of our work, and we deeply value your thoughtful engagement.

---

### Official Review · Reviewer_DtPr · 2023-08-04

**Soundness:** 4

**Excitement:**

4: Strong: This paper deepens the understanding of some phenomenon or lowers the barriers to an existing research direction.

**Missing References:**

Can the syntactic information be proceeded by chatgpt and output the corresponding response? I'm just curious, it doesn't affect the rating

**Paper Topic And Main Contributions:**

This paper proposed a data augmentation method for ASTE.
1) Their method is fine-grained, which would not reverse the sentiment label due to the alignment discriminator;
2) Their method can generate fluency text based on a fluency discriminator;
3) The performance of their method is superior.

**Reasons To Accept:**

1. The techniques sound well, like the usages of fluency discriminator and alignment discriminator;
2. In principle, the syntactic tree may boost generation diversity.
3. The paper writing is clear;

**Reasons To Reject:**

1. The accuracy of the pseudo labels may bring in some noise, how to balance the benefit and noise?
2. The discriminators have the same problem, that is the low accuracy may introduce more noise, how to mitigate this?
3. The PPO algorithm is not so easy to be effective, how do you achieve it?
4. The performance of the ppo algorithm is affected by the performance of the discriminators a lot. What kind of performance of the discriminators can make PPO effective?

**Reproducibility:**

4: Could mostly reproduce the results, but there may be some variation because of sample variance or minor variations in their interpretation of the protocol or method.

**Reviewer Confidence:**

3: Pretty sure, but there's a chance I missed something. Although I have a good feel for this area in general, I did not carefully check the paper's details, e.g., the math, experimental design, or novelty.

---

> ### Author Rebuttal · Authors · 2023-08-27
>
> We would like to express our great appreciation for your valuable comments to improve the quality of this manuscript.
>
> **Q1: How to mitigate the influence of noise from pseudo-labels on the generator and alignment discriminator?**
>
> A1: Pseudo-labeling inevitably introduces noise, which is indeed a noteworthy limitation of our approach. To mitigate the potential influence of this noise, we have devised the following strategies:
>
> -	We remove samples with invalid labels, including aspect terms and opinion terms not present in the sentence, along with label sequences that are structurally invalid.
> -	We implement further filtration based on the model's own confidence scores to exclude labels with low confidence.
>
> Based on our observations, these two steps collectively contribute to a significant reduction in noise levels. Subsequently, we discuss the effects of noise on the generator and alignment discriminator.
>
> -	Concerning the generator, it’s important to underscore that supervised training with pseudo-labels merely equips it with preliminary sentence generation capabilities. We refine label-sentence alignment through the subsequent reinforcement learning tuning, which heavily relies on the performance of the alignment discriminator.
> -	Turning to the discriminator, we evaluate its performance on the original training set and find it achieves an impressive accuracy of 89%. While not perfect, the discriminator can identify misalignment in the majority of cases. Importantly, our observations indicate that while borderline cases might exhibit some misalignment, samples with higher alignment scores rarely demonstrate such issues.
>
> In summary, we employ filtering strategies to mitigate noise impact, effectively reducing noise to an acceptable level. Furthermore, the influence of noise on our framework remains limited. A comprehensive analysis of noise will be incorporated into this manuscript, and we deeply appreciate your valuable comment.
>
> **Q2: How to make the PPO algorithm effective?**
>
> A2: Undoubtedly, achieving effectiveness with the PPO algorithm is not easy due to its training intricacies and inherent instability. To harness its effectiveness, we have employed several training strategies, including reward clipping, reward scaling, advantage normalization, and learning rate decay. Furthermore, extensive hyperparameter tuning has been conducted, involving parameters such as reward clip threshold, learning rate, kl_coef, vf_coef, lambda, and gamma. We appreciate your insights, and we will incorporate these details into the manuscript to enhance the reproducibility of our work.
>
> **Q3: Is it feasible to generate samples using ChatGPT?**
>
> A3: Utilizing ChatGPT to generate sentences based on labels and dependency trees is feasible, and GPT-4 has demonstrated promising capabilities in this regard. For simpler syntactic structures, GPT-4 can generate sentences that adhere to the requirements. However, when faced with more complex syntactic structures, the output from GPT-4 becomes less predictable and may easily yield misalignment with the given labels. Below are a few examples.
>
> Example1:
>
> -	**Label**: dish | non-tasty | negative
> -	**Dependency tree**: ROOT [det prep [pobj] cc conj [det prep [pobj]] prep [pobj [nmod]] punct]
> -	**GPT-4**: The dish was non-tasty with bland flavors and lacked any appealing presentation.
> -	**Our approach**: A non-tasty dish and a lot of it for $80.
>
> Eample2:
>
> -	**Label**: server | jerk | negative
> -	**Dependency tree**: ROOT [nsubj advmod dobj [det] cc conj [aux xcomp [advmod]] punct punct punct]
> -	**GPT-4**: The server was a jerk, not even trying.
> -	**Our approach**: I just had a jerk server and will be staying away!!!
>
> Eample3:
>
> -	**Label**: staff | unprofessional | negative ; staff | shady | negative
> -	**Dependency tree**: ROOT [prep [prep [pobj [det compound] cc conj [pobj [nmod [det nmod prep [pobj]] compound [amod]]]]] nsubj [det] aux aux attr [neg amod quantmod quantmod] punct]
> -	**GPT-4**: Amidst the unprofessional behavior of some staff members, the atmosphere turned shady and the overall experience was not at all pleasant.
> -	**Our approach**: Based on the unprofessional and shady staff at the Orlando Inn of Fashion high end resort this place should have been no more than 4 stars.
>
> Example4:
>
> -	**Label**: service | bad | negative ; manager | unprofessional | negative
> -	**Dependency tree**: ROOT [nsubj [det relcl [nsubj advmod prep [pobj]]] acomp [advmod] cc conj [nsubjpass [det compound] auxpass] punct]
> -	**GPT-4**: The service, whose manager was unprofessional, was bad and the staff, including the manager, were criticized.
> -	**Our approach**: The service we received here at teh restaurant was really bad and the manager I was assigned was unprofessional.
>
> Finally, we would like to extend my sincere gratitude for your insightful and constructive feedback once again. Your input has been instrumental in refining our work, and we greatly appreciate your thoughtful engagement.

---

### Meta-Review · Area_Chair_EyY2 · 2023-09-24

**Recommendation:** 5

**Metareview:**

Authors proposed an approach focusing on learning a generator that can directly generate new sentences by mixing labels and syntactic templates from different samples. The soundness and excitement are both 2 strong (4) and 1 good (3). Authors have responded and answered the questions raised by reviewers. This paper trains to generate samples for dataset augmentation and this method is shown to be effective. It can be helpful to the community and could be applied to other tasks.

---

### Decision · Program_Chairs · 2023-10-07

**Decision:**

Accept-Main

**Comment:**

Authors proposed an approach focusing on learning a generator that can directly generate new sentences by mixing labels and syntactic templates from different samples. The soundness and excitement are both 2 strong (4) and 1 good (3). Authors have responded and answered the questions raised by reviewers. This paper trains to generate samples for dataset augmentation and this method is shown to be effective. It can be helpful to the community and could be applied to other tasks.